# IMPROVING LANGUAGE MODEL DISTILLATION THROUGH HIDDEN STATE MATCHING

**Sayantan Dasgupta**
Computing & Information Science
University of Melbourne
Melbourne, VIC, Australia
sayandg@umich.edu

**Trevor Cohn**[*]
Computing & Information Science
University of Melbourne
Melbourne, VIC, Australia
trevor.cohn@unimelb.edu.edu

## ABSTRACT

Hidden State Matching is shown to improve knowledge distillation of language models by encouraging similarity between a student and its teacher's hidden states, as demonstrated by DistilBERT and its successors. This typically uses a cosine loss, which restricts the dimensionality of the student to the teacher's, severely limiting the compression ratio. We present an alternative technique using Centered Kernel Alignment (CKA) to match hidden states of different dimensionality, allowing for smaller students and higher compression ratios. We show the efficacy of our method using encoder–decoder (BART, mBART & T5) and encoder-only (BERT) architectures across a range of tasks from classification to summarization and translation. Our technique is competitive with the current state-of-the-art distillation methods at comparable compression rates and does not require already pretrained student models. It can scale to students smaller than the current methods, is no slower in training and inference, and is considerably more flexible. The Code is available on github[1]

## 1 INTRODUCTION

Modern LLM sizes have increased dramatically over the past few years, alongside their computational requirements. This gives rise to the need for knowledge distillation (KD) of language models with a high compression ratio, in order to produce small, fast models for inference that capture the key capabilities of learning foundation models. An $L \times D$ transformer with $L$ layers and $D$ hidden states usually has fully connected modules of dimension $D \times O(D)$, leading to a computational cost of $\mathcal{O}(D^2)$ for every layer. With slight abuse of notation, the memory required for the inference of a transformer is $\mathcal{O}(LD^2)$, motivating the need for streamlined models with smaller $D$ for downstream inference on resource-constrained devices. Xue et al. (2023) demonstrated that deeper and narrower architectures typically yield the best performance for encoder-only models. Since the encoder plays a pivotal role in Encoder-Decoder models, it provides the motivation to reduce the hidden state dimension of the teacher during compression rather than reducing only the number of layers.

Existing distillation methods use cosine loss between the hidden states, such as in DistilBERT (Sanh et al., 2019) or Shleifer & Rush (2020) on BART and mBART. This limits their application to students with the same hidden state dimensionality as the teacher, severely restricting the compression ratio. An exception is Jiao et al. (2020), which handles students with smaller dimensions using a linear projection to match the student and teacher's hidden states. This practice remains state-of-the-art and has recently been employed in Muralidharan et al. (2024). Our work aims to distill students with smaller dimensions than the teacher with a compression ratio typically $> 2\times$ using a hidden loss based on Centered Kernel Alignment (CKA - Kornblith et al. (2019)). Existing methods for sequence-level KD, such as Shleifer & Rush (2020), are limited to a compression ratio of $2\times$. However, with the size of modern LLMs going into several billions of parameters, distillation with a low compression ratio has minimal impact. Other KD approaches include aligning the student and teacher attention matrices, such as in Wang et al. (2020). We instead start with benchmarks such as

---

[*]Also at Google Research

[1]https://github.com/Sayan21/ICLR25-CKA

Sanh et al. (2019) and Shleifer & Rush (2020), which use the hidden layer loss in addition to KL Divergence and Masked or Causal modeling loss, respectively. Any gain from attention matching for our methodology will also apply to the existing benchmarks.

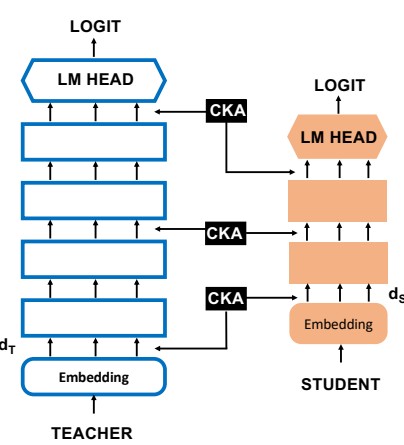

Figure 1: CKA loss between the layers of the student and the teacher. The layers with solid color are trainable.

The first attempt to solve a similar problem was DeepCCA (Andrew et al., 2013), which can align the hidden states of the student and the teacher through projection. However, DeepCCA is computationally expensive and difficult to scale when one of the dimensions is high. Instead, we use CKA to match the student and teacher's hidden states of different dimensionality and formulate a stochastic loss that can be scaled across mini-batches. This enables us to create streamlined student models with lower hidden state dimensions, which gives competitive results even from random initialization. In contrast, Sanh et al. (2019) and Shleifer & Rush (2020) achieve performance benefits by initializing the student layers with the teacher's weights, which is impossible when the student dimension is smaller.

We show that CKA is also effective in pretraining distillation for encoder–decoder models like mBART and T5 for multilingual tasks. Encoder–decoder models offer a unique advantage over decoder-only models in terms of KD: using the encoder as support, the decoder layers can be pruned to only a handful or even one layer to speed up inference (Shleifer & Rush, 2020). However, distilling the encoders in such models requires pretraining on the unsupervised corpus. Existing work such as Shleifer & Rush (2020) and Li et al. (2022) performs end-to-end distillation for machine translation. They retain the teacher's entire encoder and distill only the decoder layers. This results in a low compression ratio, with the smallest student being only half the size of the teacher. We demonstrate how pretraining distillation on multilingual corpora, utilizing a CKA-based hidden state loss, can eliminate the need to retain the teacher's encoder.

## 2 METHODOLOGY

We draw inspiration from Deep CCA in matching the hidden states of a pair of neural networks (Andrew et al., 2013). The algorithm attempts to match the representations of two networks, regardless of their dimensionality. In traditional Deep CCA, both networks are typically trained simultaneously to learn maximally correlated representations across modalities. In our adaptation, on the other hand, we keep the teacher network frozen while training the student network to match its hidden states.

Let us assume that the hidden states of the teacher and the students are $h_T \in \mathbb{R}^{d_T}$ and $h_S \in \mathbb{R}^{d_S}$ respectively, with dimensions $d_T$ and $d_S$, with $d_S \leq d_T$. Let $H_S, H_T \in \mathbb{R}^{N \times d_*}$ be the matrices with the hidden states of all the data points stacked together as rows. Canonical Correlation Analysis (CCA) takes into account the covariance and cross-correlation matrices between the hidden states, $\Sigma_{SS} = \frac{1}{N-1}\tilde{H}_S^\top \tilde{H}_S$, $\Sigma_{TT} = \frac{1}{N-1}\tilde{H}_T^\top \tilde{H}_T$ and $\Sigma_{TS} = \frac{1}{N-1}\tilde{H}_T^\top \tilde{H}_S$, where $\tilde{H}_S = H_S - \hat{\mu}_{H_S}$ and $\tilde{H}_T = H_T - \hat{\mu}_{H_T}$ are the centered hidden states of the student and the teacher with $\hat{\mu}_{H_T} = \frac{1}{N}\sum_{i=1}^{N} h_{T_i}$ and $\hat{\mu}_{H_S} = \frac{1}{N}\sum_{i=1}^{N} h_{S_i}$ as the mean of the teacher and student hidden states for $N$ samples. The goal of CCA is to learn two vectors $a \in \mathbb{R}^{d_T}$ and $b \in \mathbb{R}^{d_S}$ that maximize $R_{CCA} = \frac{a^\top \Sigma_{TS} b}{\sqrt{a^\top \Sigma_{TT} a}\sqrt{b^\top \Sigma_{SS} b}}$.

CCA is usually computed through the Singular Value Decomposition of the matrix $\Sigma_{TT}^{-1/2}\Sigma_{TS}\Sigma_{SS}^{-1/2}$ (Andrew et al., 2013). This makes the algorithm computationally very expensive. The scale of our experiments makes backpropagation with SVD unfeasible, requiring a more efficient algorithm. We adopt Centered Kernel Alignment (Kornblith et al., 2019) as an alternative to CCA for hidden state matching. Let us define $K$ and $L$ as the kernels between the hidden states of the student and the teacher respectively, as $K_{i,j} = k\left(h_{S_i}, h_{S_j}\right)$ and $L_{i,j} = l\left(h_{T_i}, h_{T_j}\right)$ for some kernel functions $k : \mathcal{H} \times \mathcal{H} \to \mathbb{R}$ and similar for $l$, with $h_{S_*}, h_{T_*} \in \mathcal{H}$ being the hidden states of

the token pairs with index $(i, j)$. Then the Hilbert Schimdt Independent Criteria is defined as $\text{HSIC}(K, L) = \text{tr}(K\Gamma L\Gamma)/(N-1)^2$, with $\Gamma$ being the centering matrix defined as $\Gamma = I - \frac{1}{N}\mathbf{1}\mathbf{1}^\top$. The authors do not note any improvement in accuracy for a non-linear kernel over a linear one (Kornblith et al., 2019). A linear kernel is also computationally less expensive, which is important for scaling the algorithm, especially for larger language models. We use a linear kernel here, for which the HSIC between the teacher and the student states is

$$\text{HSIC}(H_S, H_T) = \frac{1}{(N-1)^2}\|\tilde{H}_T^\top \tilde{H}_S\|_F^2 = \|\Sigma_{TS}\|_F^2 . \tag{1}$$

The Linear CKA between the hidden states of the teacher and the students is defined as,

$$CKA(H_S, H_T) = \frac{\text{HSIC}(H_S, H_T)}{\sqrt{\text{HSIC}(H_T, H_T)}\sqrt{\text{HSIC}(H_S, H_S)}} = \frac{\|\Sigma_{TS}\|_F^2}{\|\Sigma_{TT}\|_F\|\Sigma_{SS}\|_F} . \tag{2}$$

It can be shown that $0 \le CKA(H_S, H_T) \le 1$ (Proof in the Appendix). The authors of Kornblith et al. (2019) also show that CKA is invariant to orthogonal transforms and isotropic scaling. If the eigenvectors and eigenvalues of the covariance matrix $\Sigma_{SS}$ are $u_{S_i}$ and $\lambda_{S_i}$ respectively for $i \in [d_S]$ and similar for $\Sigma_{TT}$, then $CKA(H_S, H_T)$ can relate to them as $CKA(H_S, H_T) = \sum_{i=1}^{d_S}\sum_{j=1}^{d_T} \frac{\lambda_{S_i}\lambda_{T_j}}{\sqrt{\sum_{i=1}^{d_S}\lambda_{S_i}^2}\sqrt{\sum_{j=1}^{d_T}\lambda_{T_j}^2}}\langle u_{S_i}, u_{T_j}\rangle^2$ (Kornblith et al., 2019). Whereas if $\hat{R}_{CCA}$ is the estimated value of the CCA, it can be shown that $\hat{R}_{CCA}^2 = \frac{1}{d_S}\sum_{i=1}^{d_S}\sum_{j=1}^{d_T}\langle u_{S_i}, u_{T_j}\rangle^2$ (Kornblith et al., 2019). It can be observed that $CKA(H_S, H_T)$ turns into a quantity proportional to $\hat{R}_{CCA}^2$ when we simply replace each of $\lambda_{S_i}$ and $\lambda_{T_i}$ with a constant value. In other words, CKA is the weighted sum of the same quantities, $\langle u_{S_i}, u_{T_i}\rangle^2$, as the square of CCA with the weighting coefficient as the product of the normalized eigenvalues of the Gram matrices. From this rationale, we use the square root of CKA as a proxy for CCA to match the hidden states of the student and the teacher. The corresponding loss between the hidden states is defined as $1 - \sqrt{CKA(H_S, H_T)}$, i.e.,

$$\mathcal{L}_H = 1 - \frac{\|\Sigma_{TS}\|_F}{\sqrt{\|\Sigma_{TT}\|_F}\sqrt{\|\Sigma_{SS}\|_F}} . \tag{3}$$

## 2.1 MINIBATCH CKA

CKA, as defined above, must be computed over the entire dataset. However, it is not feasible to compute it globally over all samples. We can estimate the covariance matrices for every single minibatch, but the sample size can be very low, leading to high variance. We try to include more samples in the estimation process and compute them over $B$ mini-batches. If the covariance matrices for minibatch $b \in [B]$ are $\Sigma_{TS_b}$, $\Sigma_{TT_b}$ & $\Sigma_{SS_b}$ respectively, we can then estimate the CKA from them as the following, and then compute $\mathcal{L}_H = 1 - \sqrt{C\hat{K}A(H_S, H_T)}$, where

$$C\hat{K}A(H_S, H_T) = \frac{\|\sum_{b=1}^B \Sigma_{TS_b}\|_F^2}{\|\sum_{b=1}^B \Sigma_{TT_b}\|_F\|\sum_{b=1}^B \Sigma_{SS_b}\|_F} . \tag{4}$$

The hidden states of the transformers are accessible after the Layernorm module (Ba et al., 2016), so they have usually already been centered w.r.t. the mean of the batch. We incorporate the distillation loss between the teacher and the student probabilities ($\mathcal{L}_{Dist}$), typically defined in terms of KL Divergence (Hinton et al., 2014). We finally add a causal language modeling (CLM) loss for the student, making the final loss $\mathcal{L}_{CLM} + \mathcal{L}_{Dist} + \mathcal{L}_H$. For pre-training distillation of such LMs, for a document $X$ with $T$ tokens with $x_t$ being the one-hot vector for the token $t$, and $Y$ being the target sequence for some supervised data, the CLM losses for unsupervised and supervised cases are defined as,

$$\mathcal{L}_{CLM}(x) = -\sum_{t=1}^T x_t \log P\left(\hat{x}_t|x_{<t}\right), \qquad \mathcal{L}_{CLM}(x, y) = -\sum_{t=1}^T y_t \log P\left(\hat{y}_t|x_{<t}, y_{<t}\right) . \tag{5}$$

| Task | Teacher | #Params | Pre-training | Task-specific |
|------|---------|---------|--------------|---------------|
| Summarization | BART-large $(24 \times 1024)$ | 440M | None | CNN, XSum |
| MT | mBART-large $(24 \times 1024)$ | 610M | mC4 | EN→RO, EN→FR |
| MT with Prompt | Flan-T5-3B $(48 \times 2048)$ | 3B | mC4 | EN→ES |
| Classification | BERT-base $(12 \times 768)$ | 110M | C4 | GLUE |

Table 1: Details of the pre-training as well as supervised datasets used for different tasks

## 3 EXPERIMENTS

Here, we describe the experiments for KD with CKA for three different tasks: summarization (BART) in Section 3.1, machine translation (mBART in Section 3.2 and T5 in Section 3.3), and classification with an encoder-only model (BERT) in Section 3.4. We construct our baseline using a linear projection (Lin) to match the students' hidden state dimension to that of the teacher's, followed by an MSE loss, similar to Jiao et al. (2020). Our distillation approach follows two stages,

1. Pretraining distillation of the teacher using an unsupervised corpus (except for BART)
2. Supervised distillation (BARTs & T5) or fine-tuning (BERT) using a supervised dataset

The details of the datasets for pretraining and downstream tasks are mentioned in Table 1 for different models following the order in which they appear in this section. We keep the temperature at 1 unless mentioned otherwise and do not use hyperparameters to weigh the loss contributions. The experimental details, including learning rate, batch size, and the GPUs used, are discussed in the Appendix.

### 3.1 DISTILLATION FOR SUMMARIZATION

We begin by distilling BART-large (Lewis et al., 2020) for the downstream task of single-document news summarization, using four student architectures as shown in Table 2. We follow the experimental setup of Shleifer & Rush (2020), who perform distillation for summarization on the CNN Daily Mail (Hermann et al., 2015) and XSum (Narayan et al., 2018) datasets. For a document $x$ and its summary $y$, the supervised loss is defined as in Equation (5). The other two losses are the KL Divergence and the hidden state loss between the student and the teacher. We measure the performance using Rouge (Lin, 2004).

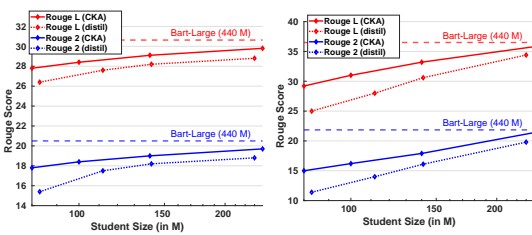

(a) ROUGE Scores - CNN  (b) ROUGE Scores - XSUM

Figure 2: ROUGE scores vs. size of the BART students trained with CKA loss, and of distilBART (Shleifer & Rush, 2020) of around the same size trained with cosine loss between the hidden layers and initialized with teacher's weights. CKA produces higher ROUGE scores.

We distill students between 6 and 24 layers and hidden states dimensions between 640 to 768 (see Table 2). We apply the CKA loss over each hidden layer of the student, applied against uniformly spaced layers in the teacher to accommodate shallower student models. We also distill the same students using linear projection-based loss between the same pair of hidden states, then with no hidden loss Table 2. We do not distill any student with the same hidden state as the teacher's (1024), as CKA would be pointless for this case. We also create distilBART models with 2, 4, 6, and 12 layers, using the same hidden dimensions as the teacher (1024), as in Shleifer & Rush (2020). These distilBART students are initialized by copying the alternate layers of the teacher and distilled with cosine loss between the hidden layers using the same hyperparameters as the other students in Table 2. When we compare the Rouge scores against the student size for these two dissimilar architectures, we see narrower students with CKA loss perform better (Figure 2). This shows that when trained well, narrower encoder–decoders outperform their wider counterparts; a similar trend is also observed in encoder-only models in (Xue et al., 2023).

| Model | P(M) | C.R. | R2(CNN) | | RL(CNN) | | R2(XSum) | | RL(XSum) | |
|-------|------|------|---------|---|---------|---|----------|---|----------|---|
| BART-large (24 x 1024) | 440 | 1.0× | 21.0 | | 30.6 | | 21.8 | | 36.5 | |
| KD wo H (6 × 640) | 80 | 5.5× | 15.1 | | 25.8 | | 13.5 | | 27.4 | |
| Lin-B (6 × 640) | 80 | 5.5× | 14.8 | −0.3 | 25.6 | −0.2 | 12.7 | −0.8 | 26.7 | −0.7 |
| CKA-B (6 × 640) | 80 | 5.5× | 16.8 | +1.7 | 26.8 | +1.0 | 15.0 | +1.5 | 29.2 | +1.8 |
| KD wo H (6 × 768) | 100 | 4.4× | 16.4 | | 26.8 | | 15.1 | | 29.2 | |
| Lin-B (6 × 768) | 100 | 4.4× | 15.5 | −0.9 | 26.2 | −0.6 | 14.1 | −1.0 | 28.2 | −1.0 |
| CKA-B (6 × 768) | 100 | 4.4× | 17.7 | +1.3 | 27.7 | +0.9 | 16.5 | +1.4 | 31.0 | +1.7 |
| KD wo H (12 × 768) | 140 | 3.1× | 17.7 | | 27.7 | | 17.6 | | 32.0 | |
| Lin-B (12 × 768) | 140 | 3.1× | 17.7 | +0.0 | 27.8 | +0.1 | 17.7 | +0.1 | 32.1 | +0.1 |
| CKA-B (12 × 768) | 140 | 3.1× | 18.5 | +0.8 | 28.5 | +0.8 | 18.7 | +1.1 | 33.5 | +1.5 |
| KD wo H (24 × 768) | 239 | 1.8× | 19.0 | | 29.1 | | 20.3 | | 34.7 | |
| Lin-B (24 × 768) | 239 | 1.8× | 19.2 | +0.2 | 29.3 | +0.2 | 20.7 | +0.4 | 35.2 | +0.5 |
| CKA-B (24 × 768) | 239 | 1.8× | **19.5** | +0.5 | **29.6** | +0.5 | **21.3** | +1.0 | **35.8** | +1.1 |

Table 2: ROUGE-2 (R2) and ROUGE-L (RL) scores for different BART students on the CNN and XSUM datasets for KD with CKA. Every BART student has an equal number of encoder and decoder layers. "KD wo H" stands for KD without a loss on the hidden states, Lin-B for KD with the linear projection-based loss, and CKA-B for CKA loss. All the students are trained with the same hyperparameters. The numbers on the right of every column of Rouge score are the differences from the baseline ("KD wo H"), in green when positive and red when negative. C.R. is the compression ratio

We further experimented with distilling the BART-large into students with smaller hidden dimensions for all the baselines and CKA, as listed in Table 2. The linear projection gives benefit up to a compression ratio of 3×, beyond which it degrades the results. The CKA method, however, improves the performance for every case when we study the ablation with respect to the hidden layer loss, and the margins of improvement increase with the compression ratio. For the highest compression ratios of 5.5×, CKA increases the Rouge score by at least 1.0, while the linear loss fails to improve the result.

## 3.2 DISTILLATION FOR MACHINE TRANSLATION

Next, we distill a multilingual mBART model (Liu et al., 2020) for machine translation. We choose deep and narrow student architectures with the settings $12 \times 384$, $12 \times 512$, $24 \times 512$, and $24 \times 640$, all having lower dimensions than the teacher (Table 3). As was the case with BART, we only consider students with smaller hidden dimensions than the teacher.

We used multilingual data from mC4 (Xue et al., 2020) for all the languages the teacher mBART model covers (details in Appendix). We used a causal modeling loss on the input (Equation (5)) and uniformly weighed the loss terms. We used a context size of $512$ and trained the students for $25$ epochs, each containing $40,000$ text samples of mC4, and computed the sum of CLM loss and KL divergence on the validation set of mC4 at the end of every epoch (Figure 3a). For the $24 \times 640$ and $24 \times 512$ models, we use the CKA loss between every pair of student and teacher hidden states. For $12 \times 512$ and $12 \times 384$, we use every alternate layer of the teacher.

The larger models converge faster, while the smaller students take much longer to converge. We plot the sum of the CLM loss and KL divergence in Figure 3a, and exclude the hidden loss since their values are incomparable. The loss converges faster than the KD with linear loss for the largest student (300M). KD with linear loss converges to a higher loss than CKA for the 173M student, whereas it does not even converge for the smallest student, 122M. We also pretrain a third set of models with no hidden loss to study ablation.

We distill the pre-trained mBART students for the downstream task of translation from English to Romanian using the WMT16 dataset (Bojar et al., 2016). We use the supervised loss defined in Equation (5) for the sentence pair $(x, y)$ where $x$ is an English sentence and $y$ is its Romanian translation. Table 3 presents the BLEU scores for EN to RO translation using different student architectures, with the teacher benchmark result taken from Shleifer & Rush (2020). We also train two DistilBART models with a compression ratio of approximately 2× using cosine loss between the

| Model | P(M) | C.R. | EN→RO | | | | EN→FR | | | |
|-------|------|------|-------|------|------|------|-------|------|------|------|
| | | | woPT | woH | Lin | CKA | woPT | woH | Lin | CKA |
| mB-L(24 x 1024) | 610 | 1.0× | | | 27.0 | | | | 40.0 | |
| mB (12 × 384) | 122 | 5.0× | 8.9 | 8.8 | 8.0 / -0.8 | 18.7 / **+9.9** | 26.3 | 34.5 | 30.9 / -3.6 | 39.2 / **+4.7** |
| mB (12 × 512) | 173 | 3.5× | 14.3 | 19.8 | 17.9 / -1.9 | 22.3 / +2.5 | 34.3 | 37.4 | 36.6 / -0.8 | 40.2 / +2.8 |
| mB (24 × 512) | 217 | 2.8× | 19.5 | 21.6 | 21.7 / +0.1 | 24.5 / +2.9 | 37.2 | 38.6 | 40.0 / +1.4 | 41.7 / +3.1 |
| mB (24 × 640) | 300 | 2.0× | 23.7 | 23.6 | 24.6 / +1.0 | **26.3** / +1.7 | 38.9 | 40.0 | 41.2 / +1.2 | **42.3** / +2.3 |
| dmB (2×1024) | 287 | 2.1× | | | 15.5 | | | | 31.5 | |
| dmB (4×1024) | 319 | 1.9× | | | 21.5 | | | | 39.3 | |

Table 3: BLEU scores for different mBART student models for EN-RO and EN-FR translation. Every student mBART has an equal number of encoder and decoder layers. **woPT** stands for KD with CKA but without pretraining. **woH** for KD with no hidden states loss, **Lin** for KD with the linear hidden loss, and **CKA** for KD with CKA loss, all with pretraining distillation on mC4. The distil-mBART (dmB) students are initialized with weights from the teacher layers and distilled using cosine loss between the hidden layers using the same hyperparameters as the rest of the mB students. The numbers on the right of the **Lin** and **CKA** columns are the differences from the baseline of KD with no hidden loss (**woH**), in green when positive and red when negative.

hidden layers, as described in Shleifer & Rush (2020). The lowest number of parameters a distilBART model can scale to is 287M, while we can easily scale down to smaller students. Smaller students make them accessible to practitioners with limited GPU resources. Furthermore, our CKA students achieve significantly better BLEU scores, even at a compression ratio of 2× than the distilBARTs.

We further distill an mBART model fine-tuned for context-aware machine translation from English to French (Sarti et al., 2024) on IWSLT2017 (Cettolo et al., 2017) with a context comprising up to 4 sentences. The authors also demonstrated that context-aware fine-tuning enhances translation accuracy even without context, and we utilize their fine-tuned mBART as a teacher for distilling translation without context. The training set used is a combination of 2 million instances randomly sampled (without replacement) from the English-French subset of the WMT14 dataset (Bojar et al., 2014) and the training samples of the IWSLT dataset (232K), resulting in a total of 2.23 million training samples. The evaluation is performed on the test set of IWSLT2017 (8.6K). Our largest student ($24 \times 640$) outperforms the teacher at a compression ratio of 2× (Table 3). The performance benefit can be attributed to the data augmentation from the WMT14 corpus. It is similar to the case of TinyBERT (Jiao et al., 2020), which also uses data augmentation during distillation and outperforms the teacher BERT-base for MNLI (Williams et al., 2017) at a compression ratio 2×. Our smallest student, with 122M parameters, produces a BLEU score only **0.8** lower than a teacher 5× larger.

| Model | EN→RO | EN→FR |
|-------|-------|-------|
| mB-Large (24 x 1024) | 312.4 | 108.0 |
| mBART (12 × 384) | 59.6 | 52.8 |
| mBART (12 × 512) | 61.9 | 56.5 |
| mBART (24 × 512) | 96.3 | 75.8 |
| mBART (24 × 640) | 102.0 | 81.2 |

Table 4: Inference time in ms for different mBART students, with the teacher at the top

When we compare the performance of CKA loss with that based on linear projection, CKA performs far better when we study the ablation w.r.t the hidden loss. The maximum gain in performance comes at the highest compression ratio. Similar to the case of BART, linear loss degrades performance at a high compression ratio of 5×. The variation of the BLEU score w.r.t the size of the students is shown in Figure 3. We further calculate the inference time of the distilled students on a 40GB A100 GPU. All the CKA students achieve substantially lower inference times (Table 4) than the teacher for EN-RO or better BLEU scores at moderately lower inference times, e.g., the 24-layer students for EN-FR.

Another area where our approach differs from Sequential KD used in Shleifer & Rush (2020) is the teacher-generated labels: it is very expensive to generate labels from the teacher through beam search. For example, it takes over 300 hours on the EN-RO dataset of 620K with an 80GB NVIDIA A100 GPU, with a FLOP count of around 161 PFLOPs. Generating teacher labels for a dataset with millions of training samples is extremely challenging, which precludes data augmentation during knowledge distillation (KD). Our pretraining-based approach requires no teacher decoding but only one expensive pre-training stage on a multilingual corpus (mC4), after which it can be fine-tuned for

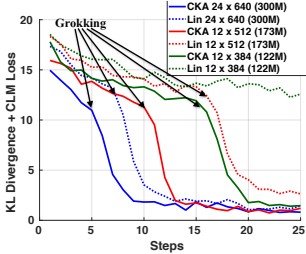 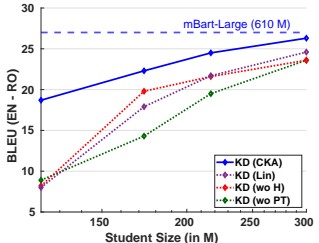 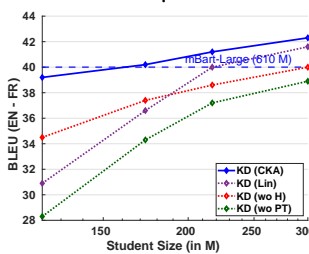

(a) Distillation Loss (mBART) for CKA and the linear loss (Lin)

(b) BLEU vs. mBART student size for EN-RO translation

(c) BLEU vs. mBART student size for EN-FR translation

Figure 3: Distillation loss for pretraining of mBART using CKA and the linear loss on the validation set of mC4 (left), with the arrows pointing to where grokking happens in the student models. BLEU score vs. parameter size of mBART for EN-RO and EN-FR translation in the next two, respectively. **wo PT** stands for KD with CKA but without pretraining, **wo H** for KD with no hidden states loss, **Lin** for KD with the hidden loss with linear projection, and **CKA** for KD with CKA loss. KD without pretraining distillation (**w/o PT**) performs the worst

specific translation tasks. For example, we use the same pre-trained models for supervised distillation in EN-RO and EN-FR. The FLOP counts for the pretraining distillation on mC4 are 79, 84, 90, and 99 PFLOPs, respectively, for the students in Table 4 in increasing order of size. In contrast, distil-mBART must repeat expensive teacher-based decoding to generate pseudo labels for every task. Our technique is thus more economical and can augment as much data as necessary to improve performance for downstream tasks, as we have done for EN-FR translation.

## 3.3 DISTILLATION OF ZERO-SHOT MODEL

Instruction-tuned language models have become the workhorse of NLP. Here, we demonstrate our technique can be applied to distill Flan-T5-3B (Chung et al., 2024), an instruction-tuned encoder–decoder model. The advantage of such models is that they can perform a wide range of tasks with reasonable accuracy without fine-tuning, which can be expensive for a 3B model. Most of the KD performed on such models in the literature is based on teacher-generated labels, as in West et al. (2022). In contrast, we perform generic KD on Flan-T5-3B, first by pretraining distillation followed by supervised KD, and skip the expensive step of generating teacher labels.

We first perform pre-training distillation on four student models: $12 \times 768$ (145M), $24 \times 768$ (T5-Base 250M), $24 \times 1024$ (425M), and $48 \times 1024$ (T5-Large 780M). We use the same mC4 corpus for pre-training, with a context length of 1024. However, since Flan-T5 is primarily trained on English tasks, we sample the English corpus of mC4 with a probability of 0.67 and add 33 other non-English language corpora, each with a probability of 0.01 (details in the Appendix). We used a context size of 1024 and trained the students for 25 epochs, each containing $40,000$ multilingual text samples from mC4 using the unsupervised loss defined in Equation (5). The experiments with CKA loss are similar to those with mBART. However, the baseline with linear projection does not converge, regardless of whether pretraining is used. Convergence is difficult for CKA loss alone, and the $12 \times 768$ and

| Model | P(M) | C.R. | FT | KD wo PT | KD wo H | CKA |
|-------|------|------|------|------|------|------|
| Flan-T5-3B ($48 \times 2048$) | 2.85B | 1.0× | 28.0 | | - | |
| T5 ($12 \times 768$) | 145M | 19.7× | 22.0 -5.2 | 23.4 -3.8 | 25.3 -1.9 | **27.2** |
| T5 ($24 \times 768$) | 250M | 11.4× | 24.3 -5.0 | 26.0 -3.3 | 27.7 -1.6 | **29.3** |
| T5 ($24 \times 1024$) | 425M | 6.7× | 26.1 -4.7 | 27.9 -2.9 | 29.4 -1.4 | **30.8** |
| T5 ($48 \times 1024$) | 780M | 3.6× | 28.0 -3.8 | 29.3 -2.5 | 30.6 -1.2 | **31.8** |

Table 5: BLEU scores for different Flan-T5 student models. **FT** stands for the BLEU of the fine-tuned Flan-T5 models without KD (zero-shot for the teacher), **wo PT** stands for KD without pretraining, **wo H** stands for KD with pretraining but no hidden states loss, and **CKA** stands for KD with pretraining using CKA loss. The numbers on the right side in the KD columns are the difference from the score for CKA to study ablation

$24 \times 1024$ models converged only after initializing the weights from the converged $24 \times 768$ and $48 \times 1024$ models.

We further run a supervised distillation on the pre-trained students for English-Spanish translation using the WMT13 corpus (Allauzen et al., 2013) by adding the prompt "Translate from English to Spanish:" in front of every English sentence. We sample 3 million sentence pairs from the WMT13 corpus, which is 14.5 million sentences in size, without replacement for training, and then measure the BLEU score on the test set. In the absence of the linear baseline, we provide the result for KD with only KL Divergence and no hidden loss in Table 5. We also fine-tune the student models on the same dataset that we use as a second baseline, and then add a third baseline for students with no pretraining distillation (Table 5). We use the Flan-T5 base ($24 \times 768$) and large ($48 \times 1024$) models from Wolf et al. (2019) for fine-tuning and create the other two models ($12 \times 768$ and $24 \times 1024$) by removing the alternate layers from them. KD with CKA-based hidden loss gives a BLEU score gain of **1.1** (for 780M) to **1.9** (for 145M) over KD with no hidden layer loss.

### 3.4 DISTILLATION OF ENCODER-ONLY MODEL

We finally apply CKA loss to the task-agnostic distillation of BERT. We discard the masked loss used in Sanh et al. (2019) and perform a pure distillation using the combination of only KL Divergence and the loss on the hidden layer, i.e., $\mathcal{L}_{Dist} + \mathcal{L}_H$. We distill the BERT-base models into student models of several configurations: $12\,\text{L} \times 512\,\text{D}$, chosen to have the same number of parameters as DistilBERT (67M); $8 \times 512$ slightly smaller than $4-$ layer DistilBERT (52M); and two smaller models ($12 \times 384$ and $6 \times 384$) with a reduced intermediate size 1536.

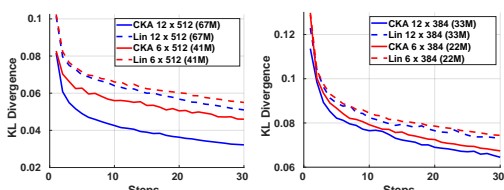

(a) KL Divergence for students with $D = 512$

(b) KL Divergence for students with $D = 384$

Figure 4: Difference in KL Divergence of KD with CKA Loss vs. the baseline of linear projection (Lin) on the validation set of C4 corpus. KD with CKA always results in a lower KL Divergence across all student sizes.

Similar to the case of mBART, we distill the student first using C4. We replace the cosine loss on the hidden layers of DistilBERT with the CKA loss. We add CKA loss between every pair of hidden states for the 12-layer student, skip every 3rd layer for the 8-layer student, and use every alternate layer for the 6-layer student. We train the model for 30 epochs, with each step involving 320, 000 sample texts from the C4 training set, and compute the KL Divergence for the C4 validation set at the end of every epoch. The KL Divergence plots are shown in Figure 4 for the CKA loss compared to the baseline method with a linear projection for various student models. CKA performs better for students of all sizes.

We further fine-tune the distilled student with CKA loss on downstream GLUE tasks, specifically: SST-2 (Socher et al., 2013) for sentiment classification; MRPC (Dolan & Brockett, 2005), QQP and STS-B for paraphrase similarity matching (Conneau & Kiela, 2018); MNLI (Williams et al., 2017), QNLI (Rajpurkar et al., 2016) and RTE (Wang et al., 2018) for natural language inference; and COLA (Warstadt et al., 2019) for linguistic acceptability. We report the Matthew correlation coefficient for COLA, the F1 score for MRPC and QQP, Spearman's rank correlation for STSB, and accuracy for the remaining datasets. CKA outperforms the linear baseline for all tasks, with the largest difference observed for COLA. We do not repeat the other baselines, as the benefits of pretraining or hidden state matching for BERT distillation are well established in works like Sanh et al. (2019) and Jiao et al. (2020).

The authors of DistilBERT initialize the students by copying the weights of the alternating layers from the teacher into the student model. However, we initialize the student with random weights due to the dimension difference. Our $12 \times 512$ model is competitive with MiniLM ($6 \times 768$) of equal size (67M) and outperforms 6-layer DistilBERT on almost every task except for SST2, where it is equivalent. Since Jung et al. (2023) shows that KD with CKA is either competitive or outperforms attention matching in MiniLM for the same student architectures, we do not repeat the same experiments. Our $8 \times 512$ model outperforms the $4-$layer DistilBERT and, for MNLI, even the $6-$layer DistilBERT.

| Task | P(M) | C.R. | COLA | SST-2 | MRPC | RTE | STSB | MNLI-m | QNLI | QQP |
| --- | --- | --- | --- | --- | --- | --- | --- | --- | --- | --- |
| # of Samples | | | 8.5K | 67.3K | 3.7K | 2.5K | 5.7K | 390K | 105K | 364K |
| BERT base (12 x 768) | 110 | 1.0× | 52.1 | 93.5 | 88.9 | 66.4 | 87.1 | 84.6 | 90.5 | 71.2 |
| LinBERT (6 x 384) | 22.0 | 5.0× | 27.0 | 89.2 | 80.4 | 52.7 | 78.2 | 80.1 | 84.9 | 68.1 |
| CKABERT (6 x 384) | 22.0 | 5.0× | 29.6 | 90.1 | 82.0 | 53.8 | 80.9 | 81.0 | 86.6 | 68.3 |
| LinBERT (12 x 384) | 33.0 | 3.3× | 41.1 | 90.2 | 83.0 | 58.4 | 81.7 | 81.1 | 85.8 | 69.2 |
| CKABERT (12 x 384) | 33.0 | 3.3× | 44.8 | 91.0 | 83.9 | 61.2 | 82.9 | 82.0 | 87.1 | 69.7 |
| DistilBERT (4 x 768) | 52.2 | 2.1× | 32.8 | 91.4 | 82.4 | 54.1 | 76.1 | 78.9 | 85.2 | 68.5 |
| LinBERT (8 x 512) | 49.8 | 2.2× | 42.7 | 90.9 | 83.8 | 55.3 | 82.3 | 82.0 | 87.9 | 69.2 |
| CKABERT (8 x 512) | 49.8 | 2.2× | **45.3** | **91.8** | **86.1** | **58.5** | **83.4** | **83.0** | **88.5** | **69.7** |
| DistilBERT (6 x 768) | 66.9 | 1.6× | 49.0 | **92.3** | 86.9 | 58.4 | 81.3 | 82.6 | 88.8 | 69.6 |
| MiniLM (6 x 768) | 66.9 | 1.6× | 49.2 | 92.0 | **88.4** | **65.1** | **85.0** | 83.0 | **90.1** | **69.9** |
| LinBERT (12 x 512) | 66.5 | 1.6× | 46.5 | 91.4 | 87.0 | 61.0 | 83.3 | 83.0 | 89.6 | 69.6 |
| CKABERT (12 x 512) | 66.5 | 1.6× | **50.2** | **92.3** | 87.8 | 63.0 | **84.9** | **88.5** | 90.0 | **70.0** |

Table 6: Results for different student encoder-only models on the GLUE test set, with the teacher BERT-base at the top. The students for CKA and Linear loss (Lin) are distilled with the same hyperparameters. The DistilBERT results are taken from Jiao et al. (2020). The results of MiniLM are generated using the model from huggingface

## 4 RELATED WORK

### 4.1 KNOWLEDGE DISTILLATION OF SEQUENCE-BASED LMS

There has been extensive work on KD for downstream classification tasks with BERT. Turc et al. (2019) demonstrated that two-stage distillation typically yields better results for transformers such as BERT (Devlin et al., 2018) or GPT-2 (Radford et al., 2019) compared to single-stage distillation on downstream tasks. The first stage involves pretraining distillation on a generic, unsupervised corpus, such as Wikipedia or the OpenWebText dataset, and the students are then further distilled using supervised datasets for various downstream tasks.

The KD literature on language models can be categorized into two main areas. The first group aims to enhance the pretraining distillation of the initial stage. For example, Turc et al. (2019) uses no loss on the hidden layers, Sanh et al. (2019) uses a cosine loss, and Wang et al. (2020) uses layerwise attention matching. Our work falls into this category. The second category uses the pre-trained models and focuses on downstream tasks. This includes Sun et al. (2019), which uses MSE loss on normalized hidden states, and Fu et al. (2021), which uses contrastive hidden state matching. However, both assume that the student's dimension is the same as the teacher's.

Generative downstream tasks, such as machine translation or summarization, are usually more complicated than classification. Early work (Kim & Rush, 2016) suggested fine-tuning the students on labels generated by the teacher. Subsequently, Shleifer & Rush (2020) combined this with the KL Divergence of the logits. Other works follow this approach, such as Li et al. (2022), which includes quantization with KD, or Wen et al. (2023), which replaces the KL Divergence with Jensen-Shannon Divergence and Total Variation Distance. Recently, reinforcement learning has been used to improve divergence, such as on-policy distillation by Agarwal et al. (2024), which utilizes a reverse KL Divergence. However, our contribution focuses on hidden state matching and will give equal benefits irrespective of the divergence between the student and the teacher. Other on-policy distillation work, such as (Gu et al., 2024), also adopts a loss based on reinforcement learning, although they generate sequences from a mixture of teacher and student distributions. The generation step is expensive and limits their efficiency. and the largest dataset they use contains 15,000 data points. Unlike Agarwal et al. (2024) and (Gu et al., 2024), which use the smaller pre-trained models of Flan-T5 or other LMs as a starting point, we derive our Flan-T5 students from scratch through pre-training distillation.

### 4.2 CENTERED KERNEL ALIGNMENT

CKA was proposed to measure the similarity between different layers of deep networks (Kornblith et al., 2019). However, it has been applied far beyond comparing layers between two similar networks, including measuring similarity between heterogeneous networks (e.g., Vision Transformers

(Dosovitskiy et al., 2020) and Resnet (Raghu et al., 2021)) and in speech (Ollerenshaw et al., 2022), where its value has been shown to follow CCA closely. Raghu et al. (2019) study CCA and CKA scores of different layers during the inner loop iteration of meta-learning (Finn et al., 2017) and show that the two metrics follow a similar trend. Saha et al. (2022) uses a similar CKA Loss for feature extraction for image classification on Tiny-Imagenet and CIFAR-100.

In NLP, CKA has been used to study the similarity between the intermediate layers of BERT (Sridhar & Sarah, 2020) and to investigate the similarity between the layers of the original and fine-tuned models for BERT-style transformer models Phang et al. (2021). Recently, Jung et al. (2023) used CKA to extract structural features from BERT during distillation. However, unlike our work, they use the standard DistillBERT ($6 \times 768$) as the student with the same dimension as the teacher and do not reduce the dimension.

# 5 CONCLUSION

We proposed a novel hidden state matching using Centered Kernel Alignment for language model distillation. We perform our experiments on a wide range of teachers from 110M BERT-base to 3B Flan-T5. Based on our experiments, we make the following key observations:

- Hidden loss using CKA almost always improves the performance for both summarization and translation. The same does not hold for the linear baseline.
- The linear loss does not work beyond a compression ratio of $3\times$ for the encoder–decoders. The generative tasks result in more complex hidden states during the decoding, and the linear projection cannot match hidden states that are too disparate.
- Pretraining distillation on a multi-lingual corpus improves the performance of machine translation even without the hidden layer loss for both mBART and Flan-T5
- The higher the complexity of a model, the more significant the performance gap between CKA and the linear baseline. The performance of the linear loss is much closer to that of CKA for simpler models, such as BERT, in classification tasks. However, unlike classification, the generative tasks for more complex encoder–decoders use CLM loss based on a sequential structure. And there, the linear loss falls short. Flan-T5 is the most complex model we distill with the highest compression ratios, for which it does not converge.

For the smallest BART student (80M), CKA produces at least $+1.0$ ROUGE score improvements. The linear baseline does not converge for the smallest 122 M student for mBART, whereas it does not converge for any model for Flan-T5. We get a BLEU score improvement of $+9.9$ for the EN-RO and $+4.7$ for EN-FR translation for the smallest mBART student (122M). For Flan-T5, the smallest student (145M) produces a BLEU score only $0.8$ lower than that of a $20\times$ larger teacher.

## 5.1 WHY MULTI-LINGUAL PRETRAINING WORKS

The largest difference using pretraining with CKA occurs in the distillation of machine translation on mBART and Flan-T5 models. Why would a similar method not also work for BART? The key difference between mBART and BART is that BART is trained exclusively on English data. The supervised datasets CNN and XSUM used for summarization are also exclusively in English and contain entire paragraphs as input prompts. As long as the input texts of CNN or XSUM are reasonably representative of all the word representations of BART, the student's encoders will learn the word features. We ran a study on pretraining BART students using C4 but did not see a significant benefit in downstream performance.

The downstream translation tasks of mBART contain only a sentence or two of a specific pair of languages, which is insufficient for extracting all the teacher's word representation features. The encoder plays the most significant role in synthesizing the word representation for encoder–decoder models, while the decoder takes the features for the input sentence from the encoder through the cross-attention. Figure 3a shows the **grokking** during the pertaining distillation, and the grokked students perform much better than their counterparts without pertaining (Table 3), with the smaller ones being worse. The smaller the student's encoder, the lower the capability to learn the teacher's complex word representation features. Other works like Agarwal et al. (2024) start with the smaller Flan-T5 models as the initial students, which are already trained on multilingual datasets.

## 6 ACKNOWLEDGEMENT

The authors thank Prof. Timothy Baldwin of the University of Melbourne/MBZUAI for his valuable feedback on different experiments and the manuscript. This research was supported by The University of Melbourne's Research Computing Services (Spartan) and the Petascale Campus Initiative.

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

## A    PROOF OF THE UPPER BOUND OF CKA

We derived that for the linear case,

$$CKA(H_S, H_T) = \frac{\|\Sigma_{TS}\|_F^2}{\|\Sigma_{TT}\|_F \|\Sigma_{SS}\|_F} \tag{6}$$

It can be observed that $\|\Sigma_{TS}\|_F^2 = \text{tr}(\tilde{H}_S \tilde{H}_S^\top \tilde{H}_T \tilde{H}_T^\top)/(N-1)^2$, where tr stands for the trace of a matrix (Equation 2 in Kornblith et al. (2019)). Now, since the Gram matrices $\tilde{H}_S \tilde{H}_S^\top$ and $\tilde{H}_T \tilde{H}_T^\top$ are both positive semi-definite, using Cauchy-Schwarz inequality for their trace, we can show that

$$
\begin{aligned}
&\frac{1}{(N-1)^2} \text{tr}[\tilde{H}_S \tilde{H}_S^\top \tilde{H}_T \tilde{H}_T^\top] \\
&\leq \frac{1}{(N-1)^2} \left( \text{tr}[(\tilde{H}_S \tilde{H}_S^\top)^2] \text{tr}[(\tilde{H}_T \tilde{H}_T^\top)^2] \right)^{1/2} \\
&= \left( \frac{1}{(N-1)^2} \text{tr}[\tilde{H}_S \tilde{H}_S^\top \tilde{H}_S \tilde{H}_S^\top] \right)^{1/2} \left( \frac{1}{(N-1)^2} \text{tr}[\tilde{H}_T \tilde{H}_T^\top \tilde{H}_T \tilde{H}_T^\top] \right)^{1/2}
\end{aligned}
\tag{7}
$$

This proves $\|\Sigma_{TS}\|_F^2 \leq \|\Sigma_{SS}\|_F \|\Sigma_{TT}\|_F$, and shows that the value of $CKA(H_S, H_T)$ is bounded above by 1. And being a positive quantity, $0 \leq CKA(H_S, H_T) \leq 1$.

## B    ADDITIONAL EXPERIMENTAL DETAIL

### B.1    SUMMARIZATION (BART)

We do not use hyperparameters to weigh the loss contributions for all the experiments. We use a batch size of 16 and sum over 8 batches for the computation of CKA and the other losses through gradient accumulation, making the effective batch size 256. We use the Adam optimizer with $\eta = 1e-4$ and weight decay $5e-4$. The context size used for the input document is 1024, while the context size for the summary is 128. All the experiments are performed on an A100 GPU with 80GB memory.

## B.2 TRANSLATION (MBART & T5)

We do not use hyperparameters to weigh the loss contributions. We use Adam Optimizer with $\eta = 3e - 5$ and weight decay $5e - 4$ for all the pretraining distillation on mC4[2]. The context size for pretraining of mBART is 512. We sample the languages with the following codes from mC4 with an equal probability: ar, cs, de, en, es, et, fi, fr, gu, hi, it, ja, kk, ko, lt, lv, my, ne, nl, ro, ru, si, tr, vi, zh, af, az, bn, fa, he, id, ka, km, mk, ml, mn, mr, pl, ps, pt, sv, sw, ta, te, th, uk, ur, xh, gl, sl.

While for Flan-T5, we use a context size of 1024. We sample the English corpus of mC4 with a probability of 0.67 and 33 other languages with a probability of 0.01 with the following codes: es, ja, fa, hi, fr, zh, bn, de, it, te, ar, pl, ta, pt, ur, gl, he, ko, th, nl, id, tr, vi, ru, sv, fi, sw, ro, lt, cs, ms, so, el.

In the downstream translation tasks for both models, we use a context size of 256 for both source and target sentences. All the experiments are performed on an A100 GPU with 80GB memory.

## B.3 CLASSIFICATION (BERT)

We use a sequence length of 512 tokens during pretraining using C4[3] and use the Adam optimizer with learning rate $\eta = 2e - 4$ and weight decay $5e - 4$. We use a batch size of 32 for gradient computation and then accumulate the gradient for 40 batches, resulting in a large batch size of 1280. This is similar to using large batch sizes in Sanh et al. (2019). The covariance matrices are averaged over the 40 batches for CKA loss computation during the pretraining and added to the final batch. We do not use hyperparameters to weigh the loss contributions. All the experiments are performed on an A40 GPU with 40GB memory.

The fine-tuning on GLUE tasks is done with the Adam optimizer with learning rate $\eta = 3e-5$ to $1e-4$ and weight decay $5e - 4$ for a batch size of 64. Since CKA loss gives a better KL Divergence than the baseline, we fine-tuned only the students distilled with CKA for the downstream tasks. We did not use any hidden state loss during fine-tuning.

---

[2]https://huggingface.co/datasets/legacy-datasets/mc4
[3]https://huggingface.co/datasets/legacy-datasets/c4

