# OpenReview forum: "Improving Language Model Distillation through Hidden State Matching"
_ICLR.cc/2025/Conference — ICLR 2025 Poster_

### Official Review · Reviewer_n8kN · 2024-11-04

**Soundness:** 3
**Presentation:** 3
**Contribution:** 2
**Rating:** 6
**Confidence:** 3

**Summary:**

The paper proposes a novel technique for knowledge distillation (KD) in LLMs using Centered Kernel Alignment (CKA) to address limitations in hidden state matching. Traditional methods rely on cosine loss, which restricts the student model's architecture to match the teacher's dimensions. This paper introduces CKA to match hidden states of different dimensionalities, enabling higher compression ratios and allowing more compact and efficient student models.

The authors tested their approach on various NLP tasks:
- Summarization: Using BART on CNN and XSum datasets, evaluated by ROUGE scores.
- Machine Translation: Tested mBART with multilingual data and evaluated on EN-RO (WMT16) and EN-FR (IWSLT2017) datasets with BLEU scores.
- Classification: Used BERT on the GLUE benchmark for classification tasks, comparing with linear projection and cosine-based baselines.
The CKA-based student models consistently outperformed models with other hidden state matching.

**Strengths:**

The method is innovative, offering an effective alternative to traditional cosine loss by using CKA for hidden state matching, which enables distillation with flexible model architectures. This approach allows for higher compression ratios and more adaptable student models, overcoming limitations of previous distillation methods.
Additionally, while most other KD methods overlook computational complexity, the authors present techniques that reduce inference times while maintaining competitive accuracy.
The performance improvement is substantial.

**Weaknesses:**

- although the authors conduct experiments across three tasks, this is relatively limited compared to other KD studies. I would encourage the authors to expand their evaluation to cover more tasks within the GLUE or SuperGLUE benchmarks for a more comprehensive analysis.
- the teacher models used in this paper are relatively small (e.g., BART, T5, and BERT), whereas the current research trend is increasingly focused on larger models (e.g., over 7 billion parameters) like LLaMA, which have stronger pre-training capabilities. It would be insightful to see whether the proposed method can sustain similar performance gains with larger models, as this would make the findings more applicable to the broader research community.
- the authors appear to primarily focus on methods involving hidden state matching. I would suggest including comparisons with other methods based on KL divergence or alternative distillation losses to provide a more thorough context for the proposed approach.

**Questions:**

n/a

---

> ### Author Response · Authors · 2024-11-22
> **Rebuttal**
>
> 1. "although the authors conduct experiments...."
>
> We added the F1 scores on QQP and accuracy for QNLI in the revised version. If we have to add the results for superGLUE, we will exceed the page limit.
>
> 2. "the teacher models used in this paper are relatively small...."
>
> We go up to the **3B** version of Flan-T5. The 7B Llama or similar decoder models are pretrained with 1T or more tokens. To create a student through pretrained distillation from scratch for such models, the student must be distilled on at least billions of tokens. For example, the recent work here uses around 30B tokens https://arxiv.org/pdf/2410.17215
>
> Compare that to our experiments, where we use around 100-125M tokens for the pretraining distillation BERT, mBART, or T5. It is easier to create baselines to show improvements on CKA for such a scale of experiments. Distilling models trained on 1T or more tokens poses a significant scaling challenge. It is impossible to train models on a new methodology and then show the improvement by creating baselines when each model has to be trained on billions of tokens. We must stick to models needing less pretraining to show methodological improvement.
>
> 3. "The authors appear to primarily focus on method......"
>
> We mostly focus on hidden state matching here, which can improve KD regardless of the divergence used. We also derived some results on the BART 6x 640 student models using reverse KL divergence, which has recently been used in Agarwal et al ICLR 2024. KD with Reverse KL Divergence shows a similar improvement when using CKA loss across the hidden states.
>
>
> |  Model | P(M) |  C.R.  | R2(CNN) | RL(CNN) | R2(XSum) | RL(XSum) |
> |:----------:|:-----:|:-------:|:----------:|:------------:|:------------:|:------------:|
> |Reverse-KL wo H (6 $\times$ 640) | 80 | 5.5$\times$ | 15.8 | 26.1 | 14.0 | 28.1|
>  |CKA-BART (RKL) (6 $\times$ 640) | 80 | 5.5$\times$ | 16.5 | 26.7 | 15.4 | 29.7|
>
>
> For translation on the mBART 12x 384 model, the  BLUE scores using Reverse KL
>  |  Model | P(M) |  EN->RO (RKL wo H) | EN->RO (RKL CKA) |  EN->FR (RKL wo H)  | EN->FR (RKL CKA) |
> |:---------:|:-----:|:-------:|:----------:|:------------:|:------------:|
> |mBART (12 $\times$ 384) | 122| 12.3 | 18.7 | 32.5 | 37.5 |

---

### Official Review · Reviewer_J69D · 2024-11-08

**Soundness:** 3
**Presentation:** 3
**Contribution:** 3
**Rating:** 8
**Confidence:** 3

**Summary:**

This paper looks at improving distillation between a teacher and student model using Centered Kernel Alignment (CKA). The main benefit of this is that it allows for the hidden dimension to be a different size than most other distillation methods used today and is much more flexible. Overall, this is a nice approach with interesting use of isotropy and math that makes it a valuable insight to the field.

Experiments were done on encoder and encoder-decoder models (BART, mBART, BERT, T5) and the results on three different NLP tasks all justified the method. It would have been nice to see an experiment on a decoder-only architecture, as well as experiments on larger versions of the models discussed, but none of these downsides are enough for me to lower my review score. I’m also not super well-versed in the distillation literature so there is a chance that there are additional baselines that should have been considered that I’m not aware of, but the current experiments show the benefit of the method.

**Strengths:**

Really interesting application of CKA to fix a problem with distillation and make it much more general.

**Weaknesses:**

Larger versions of BART or mBART for at least one experiment would have been nice.

**Questions:**

I think of distillation being useful for very large models, but the current teacher models are relatively small. What happens if you try it on a larger model?

---

> ### Author Response · Authors · 2024-11-22
> **Rebuttal**
>
> Thanks for the encouraging review. We performed experiments on Flan-T5 with **3B** parameters (so-called “XL” size), a larger encoder-decoder. It can be seen as a larger version of mBART/ BART.
>
> The decoder models are pre-trained with 1T or more tokens, and to distill them, we need to train the students on at least a billion tokens. For example, the recent work uses around 30B tokens (https://arxiv.org/pdf/2410.17215). Compare that to our experiments, where we use around 100-125M tokens for the pretraining distillation BERT, mBART, or T5. Creating multiple baselines to show improvements in CKA is easier for such a scale of experiments.
>
> The linear baseline is still the SOTA, and has recently been used in https://arxiv.org/html/2407.14679v1 (Neurips '24)

---

### Official Review · Reviewer_qCBt · 2024-11-12

**Soundness:** 2
**Presentation:** 2
**Contribution:** 2
**Rating:** 3
**Confidence:** 4

**Summary:**

- This paper presents an approach to knowledge distillation that allows matching latent representations between different dimensionalities by using centered kernel alignment (CKA) instead of cosine similarity.
- Comprehensive experiments on encoder-decoder models and masked language models confirmed that the proposed method consistently achieves higher performance compared to simple baselines, such as cases without added loss or with linear projection.

**Strengths:**

- The experiments are comprehensive, covering encoder-decoder models (BART, mBART, T5) and masked language models (BERT). The task settings range from fine-tuning on specific tasks to instruction tuning. This experimental section is valuable for readers interested in knowledge distillation in similar settings.
- The CKA used is simple, and the authors utilize a linear kernel (Line 108). This choice is expected to make the implementation easier to reproduce and facilitate scalability in training.

**Weaknesses:**

- There is limited mention of related work in knowledge distillation, making it difficult to assess the value of this study. For instance, there is no reference to research that incorporates modifications to divergence (e.g., Wen et al. ACL 2023, Gu et al. ICLR 2024) or to the distillation of causal language models, which is extensively discussed in Xu et al. (arXiv:2402.13116) and is currently of significant interest to readers.
- The tasks addressed in this study do not align with the practical use cases of modern language models. For example, summarization is often performed using causal language models today, and instruction tuning is naturally suited for causal language models rather than encoder-decoder models. While papers with significant theoretical contributions may be accepted even if their experimental settings are somewhat “toy” or “outdated,” this paper does not focus on theory (there is no theoretical section). Instead, it proposes that replacing cosine similarity with linear CKA can improve knowledge distillation in various practical settings. The value of empirical studies is considerable, and this topic is indeed appealing. However, given that the experimental setup feels somewhat outdated, it may not align well with ICLR—a leading conference for representation learning.

**Questions:**

- Linear CKA is equivalent to the RV coefficient and can be thought of as similar to Pearson correlation. Is it possible to provide any reasoning or justification for why this method is effective?
- Additionally, is there a reason why linear CKA performs better than simple regularization techniques (e.g., weight decay on the student side), including methods that adjust divergence? In practice, does it empirically outperform basic or current regularization methods?

---

> ### Author Response · Authors · 2024-11-22
> **Rebuttal**
>
> 1. "There is limited mention of related work .... currently of significant interest to readers."
>
> We did cite Wen et al n line 467, “or Wen et al. (2023), which replaces the KL Divergence with Jensen-Shannon Divergence.” We are not distilling causal language models like (Gu et al.). Nonetheless, we have added a cite to this paper in the revision (line 470). However, the works on causal modeling (arXiv:2402.13116) would be of limited utility, given the different settings and the page limit.
>
>
> 2. ``The tasks addressed in this study .... encoder-decoder models."
>
> We respectfully disagree. In summarization, works on CNN and XSUM are still benchmarked against BART, e.g. Ryu et al 2024.acl-long.319. BARTScore is widely used to evaluate the quality of the summarization by comparing the summary produced by a model with that of BART. Further, modern decoder models, such as Llama-2 or Mistral are much worse in summarization than BART. See Table 5 (page 2778) of ravaut-etal-2024-context (https://aclanthology.org/2024.acl-long.153). Llama-2-13B produces Rouge-2 scores of 14.10 and 8.61, respectively, for CNN and XSum, compared to 21.0 and 21.8 for BART. The numbers for Mistral-7B or Llama-2-7B are similar to Llama-2-7B, and are much worse than BART.  A weaker teacher will lead to worse performance on KD, and we will struggle to identify whether it is because of the lack of teacher supervision or a drawback of our algorithm. Our distilled BART student of 80M parameters produces a Rouge-2 > 15 and a Rouge-L> 26 for both CNN and XSum. Please let us know if you find a causal model with less than 100M parameters producing a similar Rouge score.
>
> Further, during the inference in a causal model, it runs a forward call on the entire prompt to generate every token. In an encoder-decoder model, the encoder runs a forward call on the prompt only once. The decoder runs a forward call only on the output (e.g., the summary) for every token. Since the prompt for summarization is usually very long because it contains the entire document, it is prohibitively slow to generate a summary in a causal model. Compared to that, an encoder-decoder model calls the repeated forward passes only on the summary during inference. The most significant purpose of KD is to improve the inference time. While we agree that LLMs are a hot topic, we do not see why distilling LLMs, especially for summarization. It would have a much higher inference time than Enc-Dec students of the same size.
>
> Similarly, instruction tuning is possible for any model with generative capabilities, and Flan-T5 performs rather well compared to its Decoder counterparts of a similar size. The MMLU score for Flan-T5-3B is 49.3, and the score on BBH is 41.0. They both are higher than those of Llama2-7B (MMLU 45.3, BBH 32.7). Overall, while LLMs are making significant advances in many tasks, this is only sometimes the case; there is still a role for encoder-decoder (and encoder-only) architectures. Further, LLMs pose significant risks, passed on during KD to the students, and rejecting a paper demanding that the LLMs must be distilled, ignoring the risks, seems a little overstretched.
>
>
> 3. "While papers with....for representation learning."
>
> Our paper's main contribution is not theoretical but methodological: CKA loss improves KD for sequence-level classification and generative tasks such as summarization and translation. We submitted our paper in the "Application" track. We focused on particular NLP tasks here and used standard SOTA teachers (as argued above). KD is significantly more complex than low-rank fine-tuning or even quantizing LLMs, and relatively little work standardizes even the Vanilla KD (w/o hidden states) for decoder models/LLMs. We cannot apply our methodology to show the improvement in such a non-standard setting. We create our model from scratch, whereas works like MiniLLM use already pre-trained models.
>
> The summarization datasets are standard and have recently been used in ravaut-etal-2024-context, ACL 2024. We have already explained how significant BART is for summarization. The mBART models are still in use, e.g. Liang et al in 2024.acl-long.588.pdf. The same goes for any work on BERT, which is still benchmarked using the GLUE dataset, such as Kim et al. 2024.emnlp-main.907.  Overall, we do not agree that these are all outdated or toy experiments
>
>
>
> 4. Q1:
>
> Our approach maximizes the correlation between the hidden states of the respective layers of the student and the teacher. The hidden states carry a lot of information, as explained in Azaria et al., 2023, findings-emnlp.68. So, it is natural that incorporating any correlation between them will improve performance.
>
>
> 5. Q2:
>
> CKA and CCA are correlation measures between a pair of multivariate random variables. This is vastly different from regularization techniques such as weight decay.

---

> > ### Comment · Reviewer_qCBt · 2024-12-03
> >
> > Thank you for your response!
> >
> > First, I now have a much clearer understanding of how MLMs and encoder-decoder models remain highly valuable in practical applications today. I appreciate your detailed explanation. Regarding the original Weakness 2 I raised, particularly the concern about whether the experimental setup aligns with current trends, I would like to withdraw this criticism. It has been an educational experience for me.
> >
> > That said, I still maintain the impression that the thoroughness of the empirical validation is insufficient, especially if the paper's main focus is to empirically evaluate a simple idea in depth (I consider simplicity to be a very strong feature! just to clarify). As Reviewer n8kN has also pointed out, the lack of exploration into causal models and the fact that the experimental results primarily compare the proposed method against "no use of the method" or relatively weak baselines (e.g., LinB) suggest significant shortcomings for an empirical study.
> >
> > Regarding Q2, I used "regularization" in what is probably its most standard sense: a term that constrains the function space. The proposed method, by imposing constraints on the student model's structure through the structure imparted by the teacher, effectively narrows the function space, correct? If so, comparative experiments against other regularization methods seem like a natural empirical extension.
> > As a side note, using independence measures for regularization also strikes me as a reasonable approach. There are papers that utilize HSIC directly, for instance: Revisiting Hilbert-Schmidt Information Bottleneck for Adversarial Robustness (NeurIPS 2021).

---

> ### Author Response · Authors · 2024-12-04
> **Response to Comment (03 Dec 2024)**
>
> Thanks for the response and the clarification; highly appreciated !!! It would also be appreciated if you could adjust the ratings (3 before) according to your withdrawal of the criticism of weakness 2. The scores of reviewer n8kN and J69D corroborate their criticism of our paper. Reviewer n8kN also asked for specific results, which we added, e.g., the two classification tasks (QNLI and QQP) on BERT in the main paper and some results on reverse KL divergence for BART/mBART. Reviewer n8kN did not hold the lack of results on LLM as a reason for rejection. This review is the odd one out, suggesting a rejection. Also, you mentioned that you have a clearer understanding of Encoder/Enc-Dec models from the discussion on the paper, something you probably lacked before. It is unfair to the other reviewers if you assign a high confidence score of 4, since both are more familiar with BERT/BART/mBART, and their distillation.
>
>
> We used standard baselines from the existing literature. The first baseline is Vanilla KD without any hidden state. The second one, i.e., the linear baseline, is the SOTA for LM distillation, which has recently been used in https://arxiv.org/html/2407.14679v1 (Neurips '24). We are unaware of further advancements in the literature that match dissimilar hidden states. We explained to reviewer n8kN that distilling the modern causal models requires around a billion tokens, and it is impossible to benchmark on various architectures when every experiment takes a billion tokens. We typically train for 100-125M for BERT, mBART, or T5, which is adequate to show the efficacy of our method. We are from academia and lack the infrastructure to distill models on billions of tokens.
>
>
> Regarding Q2, we aim to establish a correlation between the teacher and student representation. But there might be more than one interpretation of it. The paper you mentioned uses HSIC regularization to increase adversarial robustness, and our scope is limited to knowledge distillation. Going by the argument, Every KD work constrains the student model's function by constraining it to the teacher's function space. However, that does not mean we should expand the KD experiments into regularization space; neither does any KD work on LM in the literature.

---

### Meta-Review · Area_Chair_EWsA · 2024-12-24

**Metareview:**

This paper proposes a knowledge distillation approach (based on mimicking hidden representations) that replaces the traditional cosine similarity loss with a Centered Kernel Alignment (CKA) objective for matching teacher and student hidden representations. The major advantage is it can work when the dimensionalities of teacher and student are different. By leveraging CKA, the authors enable higher compression ratios and more flexible student architectures without sacrificing performance. Experiments are done on several neural models and tasks (classification, summarization, and translation). Across these settings, the proposed CKA-based method consistently outperforms or rivals common baselines, especially at high compression ratios.

Strengths
1. The proposed CKA is very flexible for distillation, with the accommodation of mismatched hidden dimensionalities.
2.The authors test the efficacy of CKA under varying compression ratios and show that it remains stable, whereas the baseline linear loss failed in many settings.

Weaknesses
1. While the paper does include experiments with a 3B-parameter Flan-T5, some reviewers note the absence of experiments on currently popular larger-scale models and casual LLMs
2.The work does not investigate deeply into theoretical groundings of why CKA might offer superior optimization properties compared to alternative objectives.

Overall, this paper provides a clear, empirically well-supported argument that CKA-based hidden state matching can alleviate the limitations of cosine-similarity-based distillation, providing higher flexibility in distillation projects. The major concern is the model's architecture choice and parameter scales in experiments. But the majority of reviewers admits its novel contribution and extensive empirical evidence. I recommend acceptance of this submission.

**Additional Comments On Reviewer Discussion:**

1. Reviewer argued that tasks and benchmarks used in the paper felt outdated. The authors demonstrated the continued relevance of encoder-decoder models (e.g., BART) with comparative metrics and cited recent works validating their choices. Although I don't completely agree with the reviewer that the setting is outdated (it is just relatively small-scale), I do hope authors could describe the limitations of their experiments clearly, especially for the missing casual LM distillation.

2. Reviewers questioned whether the method generalizes to larger models. The authors conducted additional experiments on a larger Flan-T5 model (3B parameters) and justified the infeasibility of scaling to models like Llama-7B due to resource limitations.

3. Reviewer suggested broader evaluations (e.g., SuperGLUE). The authors added results for QQP and QNLI tasks.

4. Reviewer asked for theoretical groundings behind CKA. The authors explained that CKA maximizes correlations between teacher and student hidden states, which carry substantial information, and clarified how this differs from traditional regularization.

---

### Decision · Program_Chairs · 2025-01-22

Accept (Poster)